# Unified Representation for Non-compositional and Compositional Expressions

**Ziheng Zeng and Suma Bhat**
Department of Electrical and Computer Engineering
University of Illinois Urbana-Champaign, Urbana, Illinois, USA
{zzeng13, spbhat2}@illinois.edu

## Abstract

Accurate processing of non-compositional language relies on generating good representations for such expressions. In this work, we study the representation of language non-compositionality by proposing a language model, PIER, that builds on BART and can create semantically meaningful and contextually appropriate representations for English potentially idiomatic expressions (PIEs). PIEs are characterized by their non-compositionality and contextual ambiguity in their literal and idiomatic interpretations. Via intrinsic evaluation on embedding quality and extrinsic evaluation on PIE processing and NLU tasks, we show that representations generated by PIER result in 33% higher homogeneity score for embedding clustering than BART, whereas 3.12% and 3.29% gains in accuracy and sequence accuracy for PIE sense classification and span detection compared to the state-of-the-art IE representation model, GIEA. These gains are achieved without sacrificing PIER's performance on NLU tasks (+/- 1% accuracy) compared to BART.

## 1 Introduction

Non-compositionality is a characteristic of natural language, where the meaning of the expressions cannot be deduced from its components (Baldwin and Kim, 2010). These non-compositional expressions, often referred to as being *idiomatic*, assume figurative meanings and are collectively a common occurrence appearing in nearly three out of ten sentences in English (Moon et al., 1998) across various genres (Haagsma et al., 2020) . The challenges they pose to NLP systems have been acknowledged as the classical 'pain in the neck' (Sag et al., 2002) and are recently found to impact various NLP tasks negatively, such as sentiment analysis (Liu et al., 2017; Biddle et al., 2020), dialog models (Jhamtani et al., 2021), and paraphrase generation (Zhou et al., 2021). Modern NLP systems, however, are primarily driven by the notion of compositionality, which is at the core of several system components, including tokenization (Sennrich et al., 2016; Wu et al., 2016) and the self-attention mechanism (Vaswani et al., 2017). More fundamentally, recent studies (Zeng and Bhat, 2022) reveal that the pre-trained language models (PTLMs), such as GPT-3 (Brown et al., 2020) and BART (Lewis et al., 2020), are ill-equipped to represent (and comprehend) idiomatic expressions' (IE) meanings. This is demonstrated by the lack of correspondence between the IE meanings and their embeddings; IEs with similar meanings are not close in the embedding space. Conversely, IEs close in the embedding space have a significant token or syntactic overlap. From a representation standpoint, this highlights the need for language models (LMs) to handle non-compositionality through valid representations.

Efforts to generate semantically congruent representations for IEs are now coming to the fore. For instance, GIEA (Zeng and Bhat, 2022) uses a frozen pre-trained BART that is injected with trainable adapter layers (Houlsby et al., 2019; Pfeiffer et al., 2020a) to generate IE embeddings for non-compositional expressions. With better meaning-representation correspondence, the non-compositional expert GIEA performs better than BART in downstream IE processing tasks. Yet, this advance is limited by the assumption that all IEs occur in their idiomatic sense and ignores their *contextual ambiguity* that makes them *potentially idiomatic expressions* (PIEs)–their meanings can be understood either literally or idiomatically in a context-dependent manner (Haagsma et al., 2020)[1]. For example, the PIE "behind closed doors" should be interpreted literally in *Always lock valuables behind closed doors* and idiomatically in *They avoided any publicity and made all deals behind*

---

[1]For simplicity, we hereafter refer to a sentence with an idiomatic/literal PIE as an *idiomatic/literal sentence* and their PIE embedding as *idiomatic/literal embedding*.

*closed doors*. Ideally, their representations ought to be distinct in these two contexts. However, examining the representation of 235 PIEs that are largely unrelated in their literal and idiomatic context (their literal PIE embeddings and idiomatic definitions have a mean cosine similarity of 0.0047), we notice that their representations generated by the state-of-the-art (Zeng and Bhat, 2022) exhibit a high cosine similarity between their idiomatic and literal PIE embeddings (mean cosine similarity of 0.82).

Towards addressing this discrepancy, this study extends GIEA's ability in two concrete ways. First, through semantically meaningful representations for non-compositional expressions we enable effective handling of *non-compositionality*. Second, by generating context-appropriate PIE representations distinct for idiomatic and literal PIEs we enable effective *contextual disambiguation* of PIEs. Addressing these issues involves attending to the following challenges. (1) BART and GIEA's abilities should be combined to generate good embeddings for PIE in a context-dependent manner. (2) With the self-supervised reconstruction task as the sole objective, PTLM parameters are already optimized for token reconstruction from their token embeddings. To represent PIEs, BART and GIEA's learning objectives should be revamped. To address these challenges, we propose **P**otentially **I**diomatic **E**xpression **R**epresentation generator (PIER). Inspired by AdapterFusion (Pfeiffer et al., 2021), which has been used to combine task-specific adapters, PIER generates embeddings by combining the output from each GIEA adapter layer and pre-trained BART transformer layer with an attention fusion layer serving as a routing mechanism that passes compositional or non-compositional embeddings based on the context.. It is trained under the supervision of external knowledge, e.g., IE dictionary definition and PIE senses, that helps the model to disambiguate and comprehend PIEs' literal and idiomatic meanings via a cosine-similarity based learning objective and a set of mask-infilling tasks with prompts.

Our main contributions are as follows.
(1) We propose PIER, a unified language model that combines pre-trained BART's compositional and GIEA's non-compositional representation abilities to generate semantically meaningful representations for both literal and idiomatic PIEs in a context-dependent manner.
(2) We perform an intrinsic evaluation of the result-ing IE embeddings' semantic quality by clustering them into meaning groups; the idiomatic embeddings of PIER are superior compared to those of pre-trained BART in terms of homogeneity score (+0.15); additionally, we evaluate the distinctiveness between the literal and idiomatic embeddings and found that PIER can better differentiate PIE usage and has, on an average, a +0.49 larger cosine distance for idiomatic and literal PIEs in the embedding space than GIEA.
(4) Extrinsic evaluations validate PIER's utility; PIER outperforms both BART and GIEA on two PIE processing tasks–PIE sense classification (accuracy +3.12% over GIEA and +2.67% over BART) and PIE span detection (sequence accuracy +3.29% over GIEA and +28.54% over BART).
In two classic NLU tasks of sentiment classification and paraphrase identification, PIER compares more favorably with BART than GIEA, demonstrating that its NLU capabilities do not suffer at the cost of refining its PIE representation[2].

## 2 Related Work

**Non-compositional Phrase Embedding.** Traditional methods for non-compositional phrase embedding include learning adaptive weights to combine the compositional (averaging word embeddings) and non-compositional representation of the phrase (representing phrases with single tokens) (Hashimoto and Tsuruoka, 2016; Li et al., 2018a,b). These methods cannot be adopted for contextualized embeddings. PTLMs, though producing contextualized representations, are known for their inability to handle non-compositional phrases (Zeng and Bhat, 2022; Liu and Neubig, 2022). GIEA (Zeng and Bhat, 2022), the first contextualized representation model for non-compositional phrases, efficiently adapts BART using adapter modules (Pfeiffer et al., 2020a) consisting of simple, parameter-efficient projection layers added between the trained transformer layers, to produce semantically meaningful IE embeddings in a data-efficient manner compared to LM pre-training ($\sim$60MB vs. $\sim$160GB). Despite outperforming a fine-tuned full BART model, GIEA remains challenged by PIEs' semantic ambiguity. PIER addresses this limitation through architectural modifications and additional prompt-based learning objectives as detailed in Section 3.

---

[2]The code for PIER can be found at `https://github.com/zzeng13/PIER`.

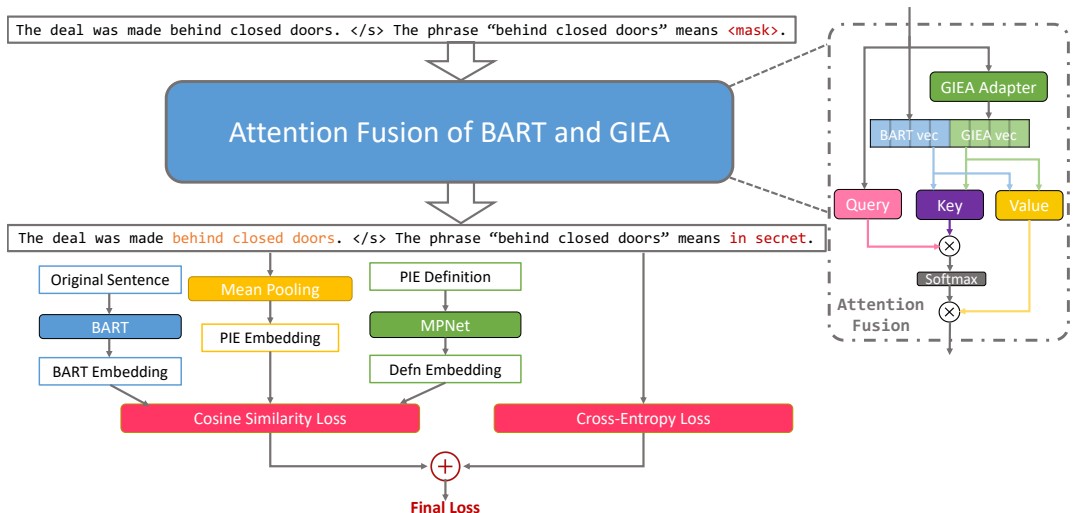

Figure 1: Overview of the PIER training framework.

**Architectures for Information Fusion.** Prior studies have explored different architectures to fuse information in neural networks. For instance, an attention flow module (Seo et al., 2017) is proposed to combine and fuse information from two vectors (query and context) for reading comprehension. Yuan and Liu (2022) infuses external graph knowledge into pre-trained BART by adding a cross-attention module inside each BART decoder layer to infuse graph entity representation. In this work, we follow GIEA and use adapters to combine and route GIEA and BART embeddings. Adapters have also shown effectiveness in multi-task and multi-lingual transfer (Pfeiffer et al., 2020b; Ansell et al., 2021). Specifically, an AdapterFusion module (Pfeiffer et al., 2021) combines multiple trained task-specific adapters with a single attention layer to automatically select appropriate adapters for a given task. In PIER, we utilize an attention fusion layer, a simplified version of an AdapterFusion module, to allow the LM to (a) combine BART and GIEA as the compositional and non-compositional language experts and (b) contextually select proper PIE representation depending on whether the PIE is used idiomatically or literally. The attention fusion layer is explained in Section 3 (See Figure 1).

**Auxiliary Guided Representation Learning.** Auxiliary information to aid learning of language representations has been explored by using phrase knowledge to mask and reconstruct token spans, e.g., noun phrases or named entities, during training to learn phrase representation (Joshi et al., 2020), and by including dictionary definitions to learn representations of rare words (Yu et al., 2022;

Zeng and Bhat, 2022). Prior work also suggests that semantically meaningful latent embeddings can be learned by optimizing the cosine similarity between source and target embeddings (Radford et al., 2021). Similarly, PIER utilizes dictionary definition for IEs to compensate for the rarity of IEs and the relatively small IE-type training instances. We also guide the PIE representation learning by optimizing the cosine similarity between the PIE embeddings and their corresponding definition/PTLM embeddings.

## 3 Unified PIE Representation Generator

To create a single language model that produces contextually appropriate embeddings for PIEs, in PIER, we combine BART's ability to generate embeddings appropriate for compositional meanings and GIEA's to non-compositional meanings such that PIER should output GIEA-style embeddings when the PIE is used idiomatically and BART's embedding otherwise.

We implement an attention layer that acts on the outputs from each frozen GIEA's adapter layer and frozen BART's transformer layer and serves as a "routing" mechanism for the compositional and non-compositional type embeddings. To train the attention layer, the overall loss is the sum of (1) a cosine similarity-based part for the embedding to encode meaning via external dictionary definitions, and (2) a reconstruction cross-entropy part that teaches the embedding of the association between the PIE senses and sentence contexts. Optimizing these two losses jointly allows the model to link PIEs' meanings to their contextual uses. The

overview of PIER framework is shown in Figure 1.

## 3.1 Attention Fusion Layer

We implement attention fusion layers to route and propagate BART or GIEA's outputs layer by layer. As shown in Figure 1, we insert an attention fusion layer after each GIEA's adapter layer and BART's transformer layer to combine them with attention weights into a single embedding vector that is sent to the next BART's transformer layer; the last attention layer outputs the final embedding vector.

Specifically, each attention layer $l$ has three trainable weight matrices, namely, Key ($\mathbf{K}_l$), Value ($\mathbf{V}_l$) and Query ($\mathbf{Q}_l$). The attention layer $l$ takes two inputs, namely, GIEA's $l$-th adapter layer output at each token position $i$, $\mathbf{g}_{l,i}$ and BART's $l$-th transformer layer output, $\mathbf{b}_{l,i}$; then, it computes the contextually attention weighted representation as

$$\mathbf{h}_{l,i} = [\mathbf{b}_{l,i}; \mathbf{g}_{l,i}]$$
$$\mathbf{a}_{l,i} = \text{softmax}(\mathbf{b}_{l,i}^\top \mathbf{Q}_l \cdot \mathbf{h}_{l,i}^\top \mathbf{K}_l)$$
$$\tilde{\mathbf{h}}_{l,i} = \mathbf{b}_{l,i}^\top \mathbf{V}_l$$
$$\mathbf{o}_{l,i} = \mathbf{a}_{l,i}^\top \tilde{\mathbf{h}}_{l,i}$$

Note that our attention fusion layer is a special, simplified case of AdapterFusion module; instead of fusing outputs from multiple adapters, our module fuses the embeddings from before (BART's transformer layer output) and after a GIEA adapter layer. Intuitively, the attention layer at each layer uses a linearly transformed BART's transformer layer output as a query to the GIEA and BART representation to determine how much of each token's BART's compositional representation needs to be substituted with GIEA's non-compositional representation based on its context. The attention weight $\mathbf{o}_{l,i}$ acts similarly to the PIE-specific weight that combines the compositional and non-compositional representations for computing PIE embeddings to adjust the balance and mixture between the compositional and non-compositional meaning in prior works (Hashimoto and Tsuruoka, 2016; Li et al., 2018a,b). But, our layer-wise attention weight is more contextualized and flexible.

With the attention fusion layer, we can train the model using the *copy objective*, where the input and output sequences are identical, just as the GIEA model does. However, our experiments later demonstrate that simply adding the attention fusion layer with the copy objective is not enough to effectively learn PIE representations. Therefore,

we have developed and incorporated the similarity learning objective and prompt infilling objectives, which we will describe in the subsequent sections.

## 3.2 Similarity Learning Objective

From prior work (Zeng and Bhat, 2022), we infer that the quality and quantity of sentences with PIE are insufficient for unsupervised representation learning. This prompts us to use dictionary definitions for idiomatic and original BART's embedding for literal PIEs to create contextual awareness.

Specifically, given a sentence with a PIE at training time, we first generate the *PIE embedding* by mean pooling PIER final output embeddings of the PIE tokens. Then, we generate two embeddings that aid the refining of the PIE embedding: (1) we generate an *idiomatic embedding* that encodes the non-compositional meaning of the PIE by using MPNet (Song et al., 2020) to produce a sentence embedding on the PIE's idiomatic dictionary definition. We use MPNet because prior work (Zeng and Bhat, 2022) found the resulting definition embeddings help representation learning more than other models such as BART; and (2) we generate a *literal embedding* that encodes the compositional meaning of the PIE by mean pooling a regular PTLM's (here, BART) final PIE token embeddings. Note that since the "literal" embeddings are contextualized, they may already encode idiomatic meanings for frequently used idioms, including idioms exclusively used figuratively. Our model should still provide accurate semantics for these idioms since the attention fusion layer could pass the idiomatic semantics through. However, for the vast majority of idioms that are rare in text, BART's embeddings are considered "compositional," not capturing their figurative meanings, hence we refer to them as *literal* embeddings.

Finally, we introduce a learning objective for sentences with a literal PIE that maximizes the cosine similarity between PIE and literal embeddings while minimizing the cosine similarity between the PIE and the idiomatic embeddings. For sentences with an idiomatic PIE, we do the opposite: encourage higher cosine similarity between PIE and idiomatic embeddings and lower cosine similarity between PIE and literal embeddings.

## 3.3 Prompt Infilling

To directly provide the PIE sense information to the model and help it relate PIE senses with sentence contexts, we design two types of prompt-based

mask infilling tasks: (1) *type classification* prompts and (2) *definition generation* prompts.

For the type classification prompts, we append the original sentence with another sentence that has a mask token, e.g., *the phrase "see red" is used in its* [MASK] *sense.*, and ask the model to infill the correct PIE sense, i.e., "idiomatic" or "literal", according to the context of the original sentence. As such, we directly inform the model of the existence and the distinction of the two PIE senses.

For the definition generation prompts, we append a masked sentence, e.g., *the phrase "see red" is used to mean* [MASK].*, and we ask the model to generate the definition of the idiomatic meaning in the place of the mask token if the PIE is idiomatic in the context; otherwise, the model should fill the mask with the PIE itself since the meaning is compositional. Through these prompts, we allow the model to learn the two PIE senses' meanings and relate them with their contexts.

We pre-defined five prompt templates for each prompt type (see Appendix A) based on our empirical observation that the variety of the prompt templates positively influences the evaluation performances (see Section 4). We append these prompts to the end of the original idiomatic or literal sentence.During training, we compute the mean cross-entropy loss for all tokens from the mask-in-filled output sentence, which we then add to the cosine similarity losses introduced in the last section to serve as the final loss. Note that unlike *prompt-based learning* (Liu et al., 2022), we use prompts to teach LMs informative representations.

# 4 Experiments

## 4.1 PIE Datasets

Similar to Zeng and Bhat (2022), we use MAGPIE (Haagsma et al., 2020), the largest-to-date dataset for English PIEs with sentences sampled from the BNC (BNC Consortium, 2007). We selected all sentences with PIEs that were unanimously labeled as idiomatic or literal by the MAGPIE annotators and have a single idiomatic definition according to Google dictionary and Wiktionary. In all, we had 32,693 sentences (77.4% idiomatic) with 1,480 PIEs in the train set and 4,102 (77.57% idiomatic) sentences with 1,001 idioms in the test set. We use MAGPIE's official random split to divide the data into train and test sets where *all* the PIEs in the test data appear in the train data. We also use idiom meaning groups proposed by Zeng and Bhat

(2022) to perform an intrinsic evaluation of the embeddings; 129 IEs form 20 groups with distinct meanings such that any two IEs from two different groups have different meanings while two IEs from the same group have similar meanings.

## 4.2 Models

We compare the performances of BART, GIEA, PIER and its variants to demonstrate the usefulness of the components in PIER.

**BART** is the pre-trained BART-base language model with six encoder and decoder layers.

**GIEA** is the non-compositional embedding generator trained with a BART-base model and adapters using the MAGPIE train set.

**BART-FT** is a BART-base model fine-tuned with the copy, similarity learning, and prompt infilling objectives.

**FusionAttn** is the model that combines BART and GIEA with the attention fusion layer and is trained with the copy objective with the cross-entropy loss.

**FusionSim** combines BART and GIEA with the attention fusion layer and is trained with the copy and similarity learning objective.

**FusionPrompt** combines BART and GIEA with the attention fusion layer and is trained with the prompt infilling objective. The above four models are used to show the usefulness of the different components of our model.

**PIER and PIER+.** PIER combines BART and GIEA with the attention fusion layer and is trained with the type classification and definition generation prompts with the reconstruction, copy, and similarity learning objectives. Only a *single* prompt template is provided for each prompt type. PIER+ is similar to PIER, but for each prompt type, we provide five templates. This model shows the benefit of using multiple prompts for each prompt type and is considered our final model.

## 4.3 Evaluation Tasks

We conduct *intrinsic* and *extrinsic* evaluation tasks.

### 4.3.1 Intrinsic Evaluation

An intrinsic evaluation indicates if PIE embeddings are semantically meaningful and distinctive in the respective literal and idiomatic contexts.

**Embedding Generation.** We evaluate the embedding quality by the competing models. We use a candidate model for each sentence to compute and mean pool the PIE token embeddings to get a single embedding vector. Then, we compute and

mean pool across all idiomatic sentences to get the idiomatic embedding for the PIE. Similarly, we get the literal embeddings for the PIEs. With the embeddings, we perform two intrinsic evaluations.

**Embedding Clustering.** The procedure is in line with Zeng and Bhat (2022). Specifically, given a model, we compute the idiomatic PIE embeddings for 129 idioms and cluster them into 20 distinct meaning groups using agglomerative clustering with complete linkage and pairwise embedding cosine similarity as the distance metric. We measure clustering quality using a *homogeneity score* and the *mean inter-group cosine distance* between the embeddings for IEs from different groups. Because the ground truth meaning groups are distinct, the larger the homogeneity scores (the score is 1.0 if all clusters contain only IEs from the same meaning group) and the mean inter-group cosine distances, the better the clustering quality.

**Embedding Differentiation.** The clustering evaluation examines only the model's ability to produce high-quality *idiomatic* embeddings. As discussed in Section 1, it is important for the language model to become innately aware of the difference between the idiomatic and literal meanings of the same PIE based on their context. Hence, given a model, we generate idiomatic and literal PIE embeddings for PIEs with both idiomatic and literal sentences from the MAGPIE test set. We compute *mean inter-type cosine similarity* between a pair of idiomatic and literal PIE embeddings across all PIEs with both literal and idiomatic sentences from the MAGPIE test set (there are 235 such PIEs). Assuming a weak correlation between the literal and idiomatic meanings for PIEs, the smaller the mean inter-type cosine similarity, the better the differentiation.

### 4.3.2 Extrinsic Evaluation

We include two classic PIE processing tasks and two NLU tasks for the extrinsic evaluation.

**PIE Sense Classification (SenseCLF)** is a classic PIE processing task (Fazly et al., 2009; Feldman and Peng, 2013; Rajani et al., 2014; Peng and Feldman, 2016; Salton et al., 2016; Liu and Hwa, 2017; Taslimipoor et al., 2018; Peng et al., 2014; Liu and Hwa, 2019), a.k.a. idiom type classification. Each sentence with a PIE is classified into two classes, *idiomatic* (positive) and *literal*, based on the PIE uses. Given a sentence with a PIE and its location, we first use the model to generate its PIE embedding; then, the PIE embedding is passed to a linear and softmax layer to perform the binary classifica-

tion. The classifier's linear layer is trained with the MAGPIE train set and is evaluated with F1 score and accuracy on the MAGPIE test set.

**PIE Span Detection (SpanDET)** is a more recent PIE processing task, a.k.a. IE identification (Zeng and Bhat, 2021; Škvorc et al., 2022), which is a special case of MWE identification Baldwin and Kim (2010) focusing on PIEs. Given a sentence with a PIE, a token-level classifier is asked to classify every token as either *idiomatic* (positive) or *literal*; when a PIE is used literally, all tokens are classified as literal; otherwise, the tokens from the PIE are labeled as idiomatic. To succeed, the classifier must correctly classify *every token* in the input sentence, effectively identifying the presence of an idiomatic PIE and precisely detecting its boundary simultaneously. Since each MAGPIE sentence annotates a single PIE, our models identify one idiomatic PIE per sentence. For the classifier, we input each token embedding generated by the tested LM to a two-layer MLP using ReLU activation, whose input dimension is the embedding dimension, and the hidden dimensions are halved after each layer. The classifier is trained with the MAGPIE train set, and only the MLP weights are trainable while the associated language model's weights are frozen. The performance is evaluated by *sequence accuracy* and *token recall*. In sequence accuracy, an instance is considered correct if and only if all the tokens in the sequence are classified correctly. To consider a model's ability to classify the sequence partially correct, we consider the token recall score by computing it for each test sequence and then averaging it across all test sequences.

To show that PIER does not sacrifice performance on NLU tasks, we consider two NLU tasks.

**Sentiment Classification (SentCLF)** classifies a given sentence into positive or negative sentiment. We use the SST2 (Socher et al., 2013) dataset and its default train and test splits (two classes) with 67,349 and 1,821 instances.

**Paraphrase Identification (ParaID)** classifies a pair of given sentences into paraphrase or non-paraphrase classes. We combine the MRPC (Dolan and Brockett, 2005) and PAWS (Zhang et al., 2019) datasets and their default train/test splits with a total of 53,069 train and 9,725 test instances.

For SentCLF and ParaID, we train a new adapter with the default Pfeiffer configuration (Pfeiffer et al., 2020a) stacked atop the testing models, making only the paraphrase classifier adapter trainable

during training. Performances are evaluated using the F1 score and accuracy.

Note that we freeze the testing language model and deliberately constrain the complexity of the classifiers to a linear layer, MLPs, or adapter layer to ensure that the performance primarily reflects the quality of the PIE embedding. See Appendix B for more details on the general setup.

## 5 Results and Analyses

**Intrinsic Evaluation.** As shown in Table 1, BART has the lowest homogeneity score (0.45) and inter-group cosine distance (0.037), indicating that the groups using the idiomatic embeddings do not correspond to those grouped by meaning. PIER+ has an absolute 0.15-point gain in both the homogeneity score and the inter-group cosine distance. While GIEA has a larger homogeneity score and inter-group cosine distance than PIER+, its inter-type cosine similarity is very high (0.82). This confirms that GIEA cannot generate contextually appropriate embeddings for idiomatic and literal PIEs and instead treats them as idiomatic in all contexts, thus ignoring their contextual ambiguity. In comparison, PIER+ achieves an inter-type cosine similarity of 0.49 lower, generating more contextually distinctive PIE embeddings. We hypothesize that PIER+ achieves a lower homogeneity score and inter-group cosine distances than GIEA because it contextually fuses BART and GIEA embeddings, thus making its idiomatic embeddings less distinctive in terms of idiomatic meanings. However, as we will show in Section 5, PIER+ embeddings encode information that helps it to achieve even better performance in PIE processing tasks. Additionally, we emphasize that PIER+ is not a mere interpolation between BART and GIEA, as it goes beyond simple combination techniques. It disambiguates idioms' literal and figurative senses based on the sentence context and generates appropriate embeddings with accurate semantics. A naïve interpolation approach, such as concatenation or averaging BART and GIEA's embeddings, would indeed result in high H-scores (0.6214 and 0.6154) and Cos-Dist (0.1503 and 0.1355), but also high DiffSim scores (0.7934 and 0.7954), which is undesirable.

**Performance on PIE Processing Tasks.** Unsurprisingly, PIER+ outperforms both BART and GIEA in the classic PIE processing tasks–SenseCLF and SpanDET–in all metrics as shown in Table 1. For the SenseCLF, PIER+ outperforms

BART by 2.66% and GIEA by 3.12%. Note that BART's type classification accuracy is already high at 93.71%, and GIEA's performance is only comparable with that of BART (not better). This is because BART and GIEA are only compositional and non-compositional expression experts (treating all PIEs as either literal or idiomatic), respectively. As shown by the intrinsic evaluation, none of their embeddings are distinctive enough for idiomatic and literal PIE embeddings. Because PIER+ produces different PIE embeddings based on context, it improves over the already high type classification accuracy from BART and GIEA.

Similarly, for SpanDET, a much more demanding task requiring detection and tagging simultaneously, PIER+ has a sequence accuracy that is 28.54% higher than BART and 3.29% higher than GIEA. We point out that 22.43% of sentences in the test set have literal PIEs, over which GIEA's sequence accuracy is only 71.84% while PIER+ has a sequence accuracy of 85.76%, gaining 13.91% absolute points in accuracy. Observing the token-level recalls, PIER+'s performance is only slightly better than that of GIEA, yet leads to a 3%+ gain in sequence accuracy. Plausibly, this is because of FusionMultiPrompt's better ability to recognize literal PIEs (as shown by the ∼14% sequence accuracy gain on literal sentences). So, with its ability to produce meaningful idiomatic embeddings, GIEA achieves high sequence accuracy in SpanDET, whereas PIER+'s ability to generate appropriate literal embeddings allows it to improve further.

**Performance on NLU tasks.** PIER+ performs competitively with BART and GIEA (F1 and accuracy differing by around +/- 1%). Given that the main purpose of PIER+ is for producing high-quality PIE embeddings and enhancing IE processing ability, the results on the NLU tasks lead us to conclude that (1) PIER+ (and GIEA to a lesser extent) adequately processes sentences with or without PIEs and thus performs comparably with BART on classic NLU tasks, i.e., PIER+ does not breakdown on sentences without PIEs; and (2) given that PIER+ produces PIE embeddings with superior semantic properties and performs very well on PIE processing tasks, we believe PIER+ is overall a better LM than BART for PIE processing.

**Effect of Individual Components.** As shown by FusionAttn's performances in Table 1 and 2, naïvely adding an attention fusion layer to combine GIEA and BART with a copy objective does

| Model | H-Score (↑) | CosDist (↑) | DiffSim (↓) |
|---|---|---|---|
| BART | 0.4546 | 0.0379 | 0.7495 |
| GIEA | **0.6450** | **0.2284** | 0.8224 |
| BART-FT | 0.4510 | 0.0331 | 0.8198 |
| FusionAttn | 0.4306 | 0.0357 | 0.8495 |
| FusionSim | 0.5015 | 0.0924 | 0.6428 |
| FusionPrompt | 0.4160 | 0.0495 | 0.7843 |
| P-Cls | 0.5756 | 0.1527 | 0.3468 |
| P-Defn | 0.5751 | 0.1546 | 0.3547 |
| PIER | 0.5844 | 0.1782 | 0.3272 |
| PIER+ | 0.6095 | 0.1838 | **0.3230** |

Table 1: Intrinsic evaluations measured by clustering homogeneity score (H-Score ↑), mean inter-group cosine distance (CosDist ↑), and mean inter-type cosine similarity (DiffSim ↓). Best performances are **boldfaced**.

| Model | SenseCLF | | SpanDET | | SentCLF | | ParaID | |
|---|---|---|---|---|---|---|---|---|
| | F1 | Acc | SA | TR | F1 | Acc | F1 | Acc |
| BART | 0.9589 | 0.9371 | 0.5076 | 0.7545 | 0.9246 | 0.9232 | 0.9165 | 0.9225 |
| GIEA | 0.9573 | 0.9325 | 0.7601 | 0.9075 | 0.9145 | 0.9117 | 0.9046 | 0.9103 |
| BART-FT | 0.9614 | 0.9408 | 0.4720 | 0.7907 | 0.9364 | 0.9346 | 0.9152 | 0.9207 |
| FusionAttn | 0.9605 | 0.9386 | 0.5651 | 0.6343 | 0.9158 | 0.9140 | 0.9031 | 0.9098 |
| FusionSim | 0.9642 | 0.9447 | 0.6131 | 0.6415 | 0.9243 | 0.9232 | 0.9068 | 0.9121 |
| FusionPrompt | 0.9632 | 0.9430 | 0.5405 | 0.8149 | 0.9025 | 0.9084 | **0.9270** | **0.9243** |
| P-Cls | 0.9712 | 0.9558 | 0.7370 | 0.8848 | 0.9208 | 0.9197 | 0.9069 | 0.9128 |
| P-Defn | 0.9720 | 0.9571 | 0.7190 | 0.8810 | 0.9315 | 0.9300 | 0.9060 | 0.9118 |
| PIER | 0.9749 | 0.9612 | 0.7864 | 0.9029 | 0.9181 | 0.9163 | 0.9027 | 0.9096 |
| PIER+ | **0.9765** | **0.9637** | **0.7930** | **0.9101** | **0.9290** | **0.9278** | 0.9068 | 0.9122 |

Table 2: Performances on extrinsic evaluation tasks; binary classification tasks, namely, PIE sense classification (SenseCLF), sentiment classification (SentCLF), and paraphrase identification (ParaID), are measured by F1 score (F1) and accuracy (Acc), and the PIE span detection (SpanDET) is measured by sequence accuracy (SA) and token-level recall (TR). Best performances are **boldfaced**.

*not* work; the homogeneity score is lower even than BART while the inter-type cosine similarity is higher than GIEA; also, FusionAttn's sequence accuracy for SpanDet is only 5.75% higher than BART yet 19.5% lower than GIEA. FusionAttn's poor performance highlights the fact that the reconstruction task with the copy objective alone is insufficient for the model to learn the intended embeddings as discussed in Section 1. Although after adding the similarity learning objective, FusionSim shows the effectiveness of similarity learning objective compared to FusionAttn ((e.g., +4.7% in SpanDET sequence accuracy), it underperforms GIEA. Similarly, FusionPrompt achieves marginal gains over BART (e.g., +3.39% in SpanDET sequence accuracy) by combining attention fusion and prompt infilling objectives, yet it severely underperforms GIEA. Moreover, having both the similarity learning and the prompt infilling objectives, BART-FT model exhibits an intrinsic quality that is similar to GIEA or BART, yet, without the attention fusion layer, it underperforms PIER+ in all intrinsic evaluation tasks and PIE processing tasks (i.e., SenseCLF and SpanDET). These results indicate that neither cosine similarity forcing nor prompt infilling objective alone is sufficient, and that it would require the utilization of the attention fusion layer to route GIEA and BART to produce appropriate PIE embeddings.

Finally, we tested the usefulness of the classification and the generation prompts through an ablation study, where we compare PIER with only the type classification prompt (P-Cls) and PIER with only the definition generation prompt (P-Defn). Each prompt type utilizes the same five prompt templates as the PIER+ model. As shown in Tables 1 and 2, even with a single prompt type, our models achieve significant improvements in both intrinsic evaluation and PIE processing tasks while maintaining a competitive performance in NLU tasks. However, when combining both prompt types, PIER+ outperforms P-Cls and P-Defn, especially in the most difficult SpanDET task, with gains of 5.6% and 7.4% in sequence accuracy, respectively. These results lead us to infer that combining the two types of prompts is beneficial and leads to further performance gains.

**Effect of Combined Components.** The salient

effect of combining all components is shown by comparing FusionSim and PIER. PIER gains 16.5% in homogeneity score and 0.32 in inter-type cosine similarity while achieving 1.65% higher accuracy in SenseCLF and 17.3% higher sequence accuracy in SpanDET. Moreover, comparing PIER+ to PIER, we observe a further meaningful gain in all metrics across all tasks, indicating the benefit of including multiple prompt templates.

### 5.1 Performance and Error Analyses

**Effect of PIE Properties.** Psycholinguistic findings shed light on how idioms' frequency, semantic and syntactic properties affect human IE comprehension (Saban-Bezalel and Mashal, 2019; Lada et al., 2023). Informed by these results, we analyze the effect of PIE training data size and IE semantic/syntactic properties on PIER's IE processing competence through correlational analyses. We found that PIE frequency in the train set does affect the PIER+'s intrinsic embedding quality but not the downstream PIE processing task performances. Additionally, we examined the correlation between PIER+'s performances from all evaluation tasks to three IE properties *decomposability*, i.e., the degree to which an IE's constituent words contribute to its figurative meaning, *literalness*, i.e., the extent an IE can be used in a literal sense, and *flexibility*, i.e., how flexible are IEs to morphosyntactic or internal modifications. We found little to no correlation between model performance and these properties. See Appendix C for more details.

**PIE Embedding Error Analysis.** Given that linguistic properties of PIEs are not the main contributing factors to PIER+'s embedding quality, we conduct further analyses on PIE embedding errors. Specifically, we analyze the PIEs that are poorly differentiated by PIER+. Selecting from the 235 PIEs used in the embedding differentiation test, we pick PIEs whose PIER+ embeddings have inter-type cosine similarity larger than 0.7 (highly similar idiomatic and literal embeddings). Observing the resulting 60 (25.5%) PIEs, we find that 44 (73.3%) have a very skewed idiomatic/literal sentence distribution in the training data. Notably, the number of idiomatic sentences divided by the number of training sentences for that PIE is either over 0.85 or lower than 0.15, suggesting that these are either only used idiomatically (idioms) or only as literal expressions in the training data, resulting in a low embedding differentiation among the PIE senses.

The remaining 16 PIEs exhibit no discernible properties, and we leave a deeper dive into these hard-to-learn PIEs for a future study.

## 6 Conclusion and Future Work

We propose PIER, a language model that uses attention fusion to combine BART and a previously proposed adapter to produce semantically meaningful and contextually appropriate representations for PIEs. Training using a prompt-infilling objective for contextual PIE-type awareness and a cosine similarity objective to guide the generated PIE embeddings toward their idiomatic or literal meanings resulted in PIER generating PIE embeddings with superior semantic properties. IE-aware PIER outperforms both BART and GIEA on IE processing tasks with BART-level performance on classic NLU tasks. These results demonstrate PIER's usefulness as an idiom-aware LM.

Future directions could explore methods to further enhance the IE awareness of broader types of non-compositional constructions beyond idioms (e.g., metaphors and similes).

## Limitations

PIER has two main limitations. First, PIER is not expected to produce embeddings for PIEs unseen during training when used in their figurative sense. This is because each PIE has a conventionalized figurative interpretation stemming from its unique origins and metaphorical linking, which requires external and PIE-specific knowledge (e.g., PIE definitions) for PIER's learning of high-quality PIE embeddings. It is likely that PIER can 'guess' the figurative meanings for certain PIEs with high decomposability or when BART already encodes their semantics during its pre-training. As such, this generalizability to unseen PIEs is not guaranteed with PIER, and we do not currently see a practical way to enable it. Second, PIER requires supervision in the form of sentences with the PIEs classified as literal or figurative. Given that BART's self-supervised learning objective during its pre-training fails to take this into account, we argue that using supervised learning with adapters is a practical alternative to capture IE semantics while reducing the number of trainable parameters and associated training data. With the availability of multi-lingual resources (Tedeschi et al., 2022) for PIEs now becoming available, and automatic PIE sense classification methods that generalize to un-

seen PIEs (Zeng and Bhat, 2021), we believe this requirement of a training corpus may be less of a bottleneck. More broadly, although we show that PIER does not lose its NLU ability on available tasks, we leave the application of PIER to IE-centered NLU tasks (e.g., NLI with figurative language) to future studies.

## Ethics Statement

The intended use of our system is to serve as a pretrained language model capable of adequately handling idiomatic language. As such, the intended users are those interested in fine-tuning PIER or using PIER's embeddings rich in idiomatic semantics for downstream NLP applications, such as detecting idiomatic expressions in text, sentiment analysis of text with idioms. In case of model failure, PIER may produce embeddings that do not accurately reflect the true figurative meanings of the IEs and thus negatively impact the downstream performance. Therefore, we advise against using PIER as part of decision models in critical situations, such as medical or financial scenarios. To ensure PIER performance as expected, idioms covered by PIER's training should be used. Beyond this, to the best of our knowledge, PIER does not introduce or contain any additional bias. PIER is trained using datasets that are publicly available and reputable. We do not collect or use any data that can violate privacy rights.

## Acknowledgements

This research was supported in part by the National Science Foundation under Grant No. IIS 2230817 and by a U.S. National Science Foundation and Institute of Education Sciences grant (2229612).

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

## A    Prompt Templates

Table 3 shows the prompt templates we used for the type classification and definition generation prompt infilling objective. During training, we randomly sample a third of the sentences to append the type classification prompts, a third to append the definition generation prompts, and the rest of the sentences are retained without appending prompts. We uniformly sample one of the five prompt templates for each chosen prompt type. We reserve sentences without masked prompts appended to ensure the model can properly generate embeddings for sentences without prompts after training. Cross-entropy and cosine similarity losses were also applied to these unaltered sentences for generating appropriate embeddings during training.

## B    Experiment Setup

We use the following implementation and checkpoints: BART-base and MPNet as maintained by Huggingface (Wolf et al., 2020) and Sentence-Transformers (Reimers and Gurevych, 2019); the adapters based on the code released by AdapterHub (Pfeiffer et al., 2020a); and GIEA as released by its authors. With the exception of BART, which requires no fine-tuning, we train all language models for 35 epochs with a batch size of 8. For PIE sense classification, all classifiers are trained for 55 epochs with a batch size of 32. For PIE span detection, all classifiers are trained for 100 epochs with a batch size of 16. For the NLU tasks, we train all classifiers for 30 epochs with a batch size of 16. We use the Adam optimizer across all experiments and set other hyperparameters to their default values.

## C    Effect of IE properties on Performances

**Effect of Training Data Size.** To understand how the availability of training data impacts PIER's performance, we compute the correlation between the PIER's (FusionMultiPrompt) per-PIE performance (averaging performances over each PIE type) to the number of training sentences for each PIE (as a proxy for their frequency in the wild). For the intrinsic evaluation, we focus on correlating each PIE's mean inter-type cosine similarity, which is indicative of the contextual appropriateness of PIER's embeddings, to its number of training instances; we find that the embedding differentiation for a

| Type Classification Prompts | Definition Generation Prompts |
|---|---|
| The phrase "[PIE]" is quite [MASK]. | The meaning of the phrase "[PIE]" is [MASK]. |
| The phrase "[PIE]" is used in its [MASK] sense. | The definition of the phrase "[PIE]" is [MASK]. |
| The phrase "[PIE]" is used as the [MASK] expression. | The phrase "[PIE]" means [MASK]. |
| The phrase "[PIE]" is the [MASK] way of saying it. | The phrase "[PIE]" is defined as [MASK]. |
| The phrase "[PIE]" takes on its [MASK] meaning. | The phrase "[PIE]" is used to express [MASK]. |

Table 3: Prompt templates for type classification and definition generation prompts. Each prompt template has a placeholder [PIE], which will be substituted into the actual PIE when appending to training sentences.

given PIE has low correlation with the data quantity for that PIE (Pearson correlation coefficient less than 0.15). However, for the PIE processing tasks, PIER's accuracy for both type classification and span detection positively correlates with the number of training sentences; the Pearson correlation coefficients and p-values are (0.73, 6e-21) and (0.66, 2e-69), respectively. So, despite the low correlation of training data size with the intrinsic embedding quality, it has a positive impact on downstream processing performance.

**Effect of IE Decomposability.** Decomposability is a semantic property of IEs capturing the degree to which an IE's words contribute to its figurative semantics. For example, *quick as a flash* is highly decomposable because its constituent words *quick* and *flash* both contribute to its overall figurative meaning, i.e., *very quickly*. On the other hand, *get your feet wet*, which is used to mean *begin to participate in an activity*, is non-decomposable. To understand the impact of IE semantic properties, we study the association (in terms of Pearson's correlation coefficient) between PIER's per-PIE performance to IE decomposability. For the quantitative decomposability scores, we use the largest-to-date linguistic resource available (Bulkes and Tanner, 2017), which contains descriptive norms for 870 American English idioms gathered from 2,100 human participants (several idioms have 100+ annotations). Of these, 39 overlap with the PIE types from the intrinsic evaluation set and 178 with the PIE types from the PIE processing tasks. The decomposability scores are binary, with 1 being decomposable and 0 being non-decomposable and were averaged over more than 100 human-assigned scores per PIE.

For the intrinsic evaluation, we find that the mean inter-type cosine similarity has a low association with PIE's decomposability (correlation of 0.2012 with a p-value of 0.2194). Similarly, for PIE sense classification, the per-PIE accuracy is also not correlated with decomposability (the magnitude of the correlation is less than 0.1). On the other hand,

for PIE span detection, the per-PIE sequence accuracy is weak but positively correlated with decomposability (correlation = 0.2537, p-value = 6e-4). Hence, PIE decomposability only weakly impacts the PIE span detection performance while having no significant impact on the intrinsic quality of PIE embeddings or PIE sense classification.

**Effect of PIE Literalness.** Here, we study the impact of PIEs' contextual ambiguity (represented as their literalness scores from Bulkes and Tanner (2017)) on PIER's performance. Literalness captures the extent to which a phrase can be used in its literal sense on a scale of 1 to 5, with a 5 indicating IE has a clear, well-formed, and plausible literal interpretation. We hypothesize that literalness would not affect the embedding quality and downstream performances since PIER is designed to address PIEs' contextual ambiguity. We use the same PIE types and per-PIE performance metrics from the decomposability study above. Computing the Pearson correlation between the literalness and PIER's performance for the intrinsic and PIE span detection tasks, we again found both correlation coefficients to be negligible with magnitudes less than 0.1. For PIE sense classification, the correlation coefficient between per-PIE accuracy and literalness is -0.1213 with a p-value of 0.1068 (not statistically significant at $\alpha = 0.05$). This suggests that PIE's literalness has no overall impact on PIER's PIE embedding and IE processing abilities.

**Effect of PIE Flexibility.** We examine the impact of PIE's syntactic flexibility (or frozenness) (Constant et al., 2017; Gehrke and McNally, 2019) on PIER's performance. We analyze the PIE span detection performance with respect to the idiom fixedness levels. According to the definitions given by Sag et al. (2002), PIEs (and lexicalized phrases in general) can be categorized into three levels: (1) *fixed* (e.g., *ahead of the game*)—PIEs with no morphosyntactic or internal modification, (2) *semi-fixed*, (e.g., *go down the rabbit hole*)—allow restricted lexical variations (*went down the rabbit hole*) such as inflection and determiner selection,

while maintaining strict word ordering and composition, and (3) *syntactically-flexible* (e.g., *stab someone in the back*)—maintain only the basic word order while allowing a multitude of syntactic variations on the internal words. We use the flexibility measurements provided by Zeng and Bhat (2021), which contains 139 PIE types. Intersecting with PIE types in our experiments, we find 60 PIE types for the intrinsic evaluation and 129 PIE types for PIE processing tasks. After computing the Pearson correlation for PIER's performance from each evaluation task, we found all correlation coefficients have magnitudes of less than 0.1, indicating that the PIE flexibility does not affect the quality of PIER's PIE embedding, nor does it impact the performance of the downstream processing task.