# OpenReview forum: "Unified Representation for Non-compositional and Compositional Expressions"
_EMNLP/2023/Conference — EMNLP 2023 Findings_

### Official Review · Reviewer_ZgHk · 2023-08-02

**Typos Grammar Style And Presentation Improvements:** None.
**Soundness:** 2

**Excitement:**

3: Ambivalent: It has merits (e.g., it reports state-of-the-art results, the idea is nice), but there are key weaknesses (e.g., it describes incremental work), and it can significantly benefit from another round of revision. However, I won't object to accepting it if my co-reviewers champion it.

**Missing References:**

None.

**Paper Topic And Main Contributions:**

This paper proposes PIER and PIER+, which aim to improve the representations for potentially idiomatic expressions (PIEs). They provide two main components: attention fusion layers, similarity learning objective, and prompt infilling. They also report the experimental results of intrinsic and extrinsic evaluation, which show the potential of PIER and PIER+ to represent PIEs.

**Questions For The Authors:**

- Please see above.

**Reasons To Accept:**

- Interesting research direction.

**Reasons To Reject:**

- The current baselines cannot show the effectiveness of attention fusion layers. All FusionAttn, FusionSim, FusionPrompt, PIER, and PIER+ have the attention fusion layers. My hypothesis is that fine-tuning BART with prompt infilling + similarity learning objective is enough. This can be also observed from Table 2, which FusionAttn does not give us any improvement. The authors should design more baselines to figure out the necessary of  attention fusion layers.
- The results of SpanDET is a bit weird. Using only FusionSim or FusionPrompt is worse than GIEA. However, Using PEIR (which is kind of FusionSim+Prompt) can perform better than GIEA. Can authors provide some explanations?
- It's worth to have some ablation studies for type classification prompts and definition generation prompts.

**Reproducibility:**

3: Could reproduce the results with some difficulty. The settings of parameters are underspecified or subjectively determined; the training/evaluation data are not widely available.

**Reviewer Confidence:**

4: Quite sure. I tried to check the important points carefully. It's unlikely, though conceivable, that I missed something that should affect my ratings.

---

> ### Author Rebuttal · Authors · 2023-08-28
>
> ***Question**: The current baselines cannot show the effectiveness of attention fusion layers. All FusionAttn, FusionSim, FusionPrompt, PIER, and PIER+ have the attention fusion layers. My hypothesis is that fine-tuning BART with prompt infilling + similarity learning objective is enough. This can be also observed from Table 2, which FusionAttn does not give us any improvement. The authors should design more baselines to figure out the necessary of attention fusion layers.*
>
> Thank you for pointing this out. We agree that applying only similarity learning and prompt infilling objectives to fine-tune a BART model without the attention fusion layer is an important and informative baseline. Consequently, we have added this as a new baseline model, named "BART-FT," to our experimental results, which we will include in the revised version of our paper.
>
> As shown in the augmented tables at the end, the BART-FT model exhibits similar intrinsic quality to GIEA or BART and underperforms PIER+ in all intrinsic evaluation tasks and PIE processing tasks (i.e., SenseCLF and SpanDET). However, it performs competitively with all other models in the NLU tasks (i.e., SentCLF and ParaID). This result indicates that our core architectural design, which involves applying attention fusion to contextually combine BART and GIEA's embeddings, is essential. Simply fine-tuning BART with auxiliary learning objectives cannot produce PIE embeddings with accurate semantic meanings and strong PIE processing ability.
>
> In other words, although FusionAttn alone is not sufficient to provide improvements, the combination of attention fusion and our proposed learning objectives leads to significant enhancements.
>
> -----
>
> ***Question**: The results of SpanDET is a bit weird. Using only FusionSim or FusionPrompt is worse than GIEA. However, Using PEIR (which is kind of FusionSim+Prompt) can perform better than GIEA. Can authors provide some explanations?*
>
> Through our experiments, we have discovered that all three components introduced in our paper – the attention fusion layer, similarity learning objective, and prompt infilling objective – are essential for generating semantically accurate representations for PIEs. Therefore, relying solely on prompt infilling or the similarity learning objective is insufficient for learning embeddings with high intrinsic quality, which in turn would be beneficial for downstream PIE processing ability.
>
> In addition to span detection, we observe a similar trend in other intrinsic and PIE processing applications, where the model achieves significant performance improvement only when all three components are combined. Consequently, the results from the span detection are consistent with the findings from our other experiments.
>
> -----
>
> ***Question**: It's worth having some ablation studies for type classification prompts and definition generation prompts.*
>
> Thank you for the suggestion. In addition to the ablation study on the number of prompts, we have included an ablation study comparing PIER with only the type classification prompt (P-Cls) and PIER with only the definition generation prompt (P-Defn). Each prompt type utilizes the same five prompt templates as the PIER+ model.
>
> As demonstrated in the table below, our model achieves significant improvements in both intrinsic evaluation and PIE processing tasks when using either type of prompts, while maintaining competitive performance in NLU tasks. However, when combining both prompt types, PIER+ outperforms P-Cls and P-Defn in intrinsic evaluation metrics, particularly in the SpanDET task, with gains of 5.6% and 7.4% in sequence accuracy, respectively. These results support the conclusion that combining the two types of prompts is beneficial and leads to further performance improvements.
>
> Furthermore, when comparing P-Cls and P-Defn, their performances are mostly competitive, with no clear winner. Thus, we conclude that both prompt types are approximately equally beneficial, and their advantages are amplified when used together.
>
>
> **Intrinsic Evaluation**
>
> | **Model** | **H-Score (↑)** | **CosDist (↑)** | **DiffSim (↓)** |
> |-----------|-----------------|-----------------|-----------------|
> | BART      | 0.4546          | 0.0379          | 0.7495          |
> | GIEA      | 0.645           | 0.2284          | 0.8224          |
> | BART-FT   | 0.451           | 0.0331          | 0.8198          |
> | P-Cls     | 0.5756          | 0.1527          | 0.3468          |
> | P-Defn    | 0.5751          | 0.1546          | 0.3547          |
> | PIER+     | 0.6095          | 0.1838          | 0.323           |
>
> **Extrinsic Evaluation**
>
> | **Model**   | **SenseCLF** |        | **SpanDET** |        | **SentCLF** |        | **ParaID** |        |
> |---------|----------|--------|---------|--------|---------|--------|--------|--------|
> |         | F1       | Acc    | SA      | TR     | F1      | Acc    | F1     | Acc    |
> | BART    | 0.9589   | 0.9371 | 0.5076  | 0.7545 | 0.9246  | 0.9232 | 0.9165 | 0.9225 |
> | GIEA    | 0.9573   | 0.9325 | 0.7601  | 0.9075 | 0.9145  | 0.9117 | 0.9046 | 0.9103 |
> | BART-FT | 0.9614   | 0.9408 | 0.472   | 0.7907 | 0.9364  | 0.9346 | 0.9152 | 0.9207 |
> | P-Cls   | 0.9712   | 0.9558 | 0.737   | 0.8848 | 0.9208  | 0.9197 | 0.9069 | 0.9128 |
> | P-Defn  | 0.972    | 0.9571 | 0.719   | 0.881  | 0.9315  | 0.93   | 0.906  | 0.9118 |
> | PIER+   | 0.9765   | 0.9637 | 0.793   | 0.9101 | 0.929   | 0.9278 | 0.9068 | 0.9122 |

---

### Official Review · Reviewer_uedZ · 2023-08-03

**Soundness:** 4

**Excitement:**

3: Ambivalent: It has merits (e.g., it reports state-of-the-art results, the idea is nice), but there are key weaknesses (e.g., it describes incremental work), and it can significantly benefit from another round of revision. However, I won't object to accepting it if my co-reviewers champion it.

**Paper Topic And Main Contributions:**

This paper proposes PIER, a model that fuses representations from BART and GIEA (a model specialised in creating idiomatic representations and makes use of adapters in order to do so), to maintain performance on compositional data (both literal PIEs and compositional data in NLU tasks), while improving representations for non-compositional phrases (specifically, idioms included by Haagsgma in the MAGPIE dataset). Fusing representations is achieved using an Attention Fusion layer that, for each layer individually, mixes the two representations using layer-specific attention weights. The model is trained using a loss consisting of
1. cosine distances that encourage similarity between idiomatic PIEs and the idiom’s dictionary definition, encourage similarity between literal PIEs and compositional representations, discourage similarity between literal PIEs and the idiom’s dictionary definition, and discourage similarity between idiomatic PIEs and compositional representations.
2. a reconstruction loss that is meant to strengthen the association between the context and the PIE

The data is presented via a prompt infilling task (where the model indicates whether a phrase is used idiomatically or literally) and a definition generation task, e.g. “the phrase `see red’ is used to mean …”, where the model should output the literal meaning or the idiomatic interpretation. The mean cross-entropy loss computed over these “definitions” serves as the reconstruction loss mentioned above.

The models trained are BART, GIEA, fusing combined with the reconstruction loss, fusion combined with the reconstruction and similarity losses, fusion combined with the prompt infilling task, full PIER, and PIER+ (that uses multiple prompts per input). The models are trained using MAGPIE data.

The evaluation consists of intrinsic and extrinsic evaluation:
1. The intrinsic evaluation measures the quality of clusters of idioms with related meanings, and measures the correlation between literal and idiomatic occurrences of a PIE.
2. Extrinsic evaluation using PIE processing tasks, specifically PIE sense classification and span detection.
3. Extrinsic evaluation using sentiment classification and paraphrase identification.

PIER outperforms GIEA on most of the extrinsic evaluation tasks, while GIEA is better from an intrinsic point of view. Note that the versions of PIER that don’t include all components (apart from the 5 prompts) typically underperform, full PIER is needed to obtain the improvements.


**Questions For The Authors:**

- line 303: when you generate a literal embedding, do you use token embeddings (layer 0) or contextualised embeddings (final layer)? I was thrown off by the usage of the word “final” in line 306, hence the question. When using the contextualised embeddings, this may, after all, not be a compositional embedding, considering that BART would have some accurate idiomatic representations, particularly for the more frequent idioms (whose idiomatic meaning is highly conventionalised).
- line 367 “...where all the PIEs in the test data appear in the train data”: have you run some experiments where that was not the case? I’m curious to see whether there can be non-iid improvements as well. I consider it somewhat unlikely, since without having seen training data, the meaning of an idiom can be somewhat unpredictable, but still.
- For the training techniques that are not specific to fusing the two models (copy objective, infilling, cosine similarity objective) have you applied those to BART or GIEA without including the fusion attention layer? That could help to disentangle your various contributions, because running full PIER might be impractical for future work, whereas adding one of those components might be quite a bit easier. (e.g. I can imagine that the copy objective in itself could help BART improve)


**Reasons To Accept:**

- PIER is a carefully crafted model that, across the board, outperforms its main competitor, GIEA. This model will be of great interest to the subcommunity of *ACL conferences that are interested in figurative and idiomatic language modelling or modelling multi-word expressions, more generally.
- Apart from full PIER, the paper presents a range of techniques that themselves can be more widely applied, even outside of the context of PIER, specifically, such as the definition generation prompting setup, or the cosine similarity losses introduced.
- The paper presents a very thorough analysis via the various intrinsic and extrinsic evaluations, highlighting both where PIER does and does not outperform previous models.

**Reasons To Reject:**

- The model is presented as one that both preserves strong compositional representations while also improving on capabilities to disambiguate between idiomatic usage of PIEs and literal usage of PIEs. However, in order to achieve that, all of the different components of PIER (multiple losses, merging representations via the attention fusion, multiple prompting techniques combined during training) need to be combined. This limits its potential to be applied outside of figurative language processing tasks: it would require quite a lot of engineering to maintain the performance on idioms while also improving on other downstream tasks.
- It is unclear how the individual components contribute to the overall performance, since all variants use the attention fusion. Potentially, a subset of the techniques but without the attention fusion could already improve GIEA and/or BART.

**Reproducibility:**

3: Could reproduce the results with some difficulty. The settings of parameters are underspecified or subjectively determined; the training/evaluation data are not widely available.

**Reviewer Confidence:**

3: Pretty sure, but there's a chance I missed something. Although I have a good feel for this area in general, I did not carefully check the paper's details, e.g., the math, experimental design, or novelty.

**Typos Grammar Style And Presentation Improvements:**

- In the equations in section 3.1 it looks like the value vectors are only bart-based (h_tilde) whereas in the figure it looks like the value vectors are both bart- and giea-based (h). Could you double-check that and revise if needed? (Unless I misunderstood something)
- line 141: Wait... where is contribution number 3?
- line 241/242: rephrase, “teaches” is very vague
- line 87 - 89: the exact cosine similarity here is a bit meaningless without a baseline... all sentence / phrase embeddings tend to have high similarity anyways, and it’s hard to tell whether 0.82 is relatively good/bad due to an issue known as the high anisotropy problem or the representation degeneration problem: LMs contextualised embeddings all end up in a small portion of the hidden space and are, therefore, more similar than one would expect. (hence the need for a baseline)

---

> ### Author Rebuttal · Authors · 2023-08-28
>
> ***Questions**: The model is presented as one that preserves strong compositional representations while also improving on capabilities to disambiguate between idiomatic usage of PIEs and literal usage of PIEs. However, to achieve that, all of the different components of PIER (multiple losses, merging representations via attention fusion, and multiple prompting techniques combined during training) must be combined. This limits its potential to be applied outside of figurative language processing tasks: it would require quite a lot of engineering to maintain the performance on idioms while also improving on other downstream tasks.*
>
> We appreciate your observation that obtaining high-quality PIE representations requires the combination of all three components introduced in this paper: attention fusion, similarity learning, and multiple prompt infilling techniques, as supported by our experimental results and ablation study. We would like to emphasize that the primary objective of this paper is to enable pretrained language models to generate high-quality PIE representations and enhance their PIE processing capabilities while preserving pretrained models’ original proficiency in non-idiom specific NLU tasks, such as sentiment classification and paraphrase identification.
>
> Achieving this goal, which entails developing a BART-based model capable of producing superior PIE embeddings, does not preclude further fine-tuning for downstream applications and, therefore, does not limit the model's potential applications to other tasks.
>
> The question of whether our model can maintain its PIE processing ability after fine-tuning is an interesting yet unexplored empirical question, as it is expected that a language model's embeddings may shift to achieve high performance for a specific downstream task after fine-tuning. Addressing this question is somewhat beyond the scope of this paper, as our primary aim is to improve the representation of idiomatic expressions and demonstrate their impact on PIE processing tasks. We have also shown that, in doing so, we did not compromise the language model's NLU capabilities. Nevertheless, investigating the changes in representation after fine-tuning in general could be a valuable direction for future research.
>
> -------
>
> ***Question**: line 303: when you generate a literal embedding, do you use token embeddings (layer 0) or contextualised embeddings (final layer)? I was thrown off by the usage of the word “final” in line 306, hence the question. When using the contextualised embeddings, this may, after all, not be a compositional embedding, considering that BART would have some accurate idiomatic representations, particularly for the more frequent idioms (whose idiomatic meaning is highly conventionalised).*
>
> Thank you for raising this point. To clarify, we utilized the output from the final layer when generating the literal embeddings, so they are indeed contextualized. It is possible that for the most frequent idioms in the pretraining corpus, the embeddings already encode their idiomatic meanings. This could also be true for any idioms that frequently appear in text and are used exclusively in their figurative sense (i.e., are highly conventionalized).
>
> However, we argue that for these idioms, our method would not negatively impact their semantics, as both GIEA and BART already contain their figurative meanings, and they appear in sentences solely in their figurative sense. Consequently, the output of our proposed PIER model should remain semantically accurate for these idioms. Moreover, for most idioms that are individually rare in text, the BART embeddings encode only their literal meanings. This is supported by the intrinsic evaluation in our paper and was also demonstrated in the original GIEA paper, where the author presented a case study showing that in BART's embedding space, idiomatic expressions are grouped by their token or syntactic structure overlaps, rather than their semantic affinities. Therefore, we generally consider BART's embeddings to be "compositional," as they do not capture the non-compositional, figurative meanings of idioms.
>
> Nonetheless, to avoid potential confusion and enhance clarity, we will provide additional clarification regarding the use of BART's embeddings and the possible cases where BART already encodes the figurative meanings for frequent idioms in our revision.
>
> -----
>
> ***Question**: [4] line 367 “...where all the PIEs in the test data appear in the train data”: have you run some experiments where that was not the case? I’m curious to see whether there can be non-iid improvements as well. I consider it somewhat unlikely, since without having seen training data, the meaning of an idiom can be somewhat unpredictable, but still.*
>
> Thank you for your comment. We acknowledge that our proposed PIER model, like GIEA, is limited to working with PIEs that are encountered during training. However, we contend that this constraint is necessary and the most feasible approach to ensure high-quality representations for PIEs when using smaller-scale pretrained language models that have not adequately learned the figurative meanings of PIEs during their pre-training phase.
>
> The main reason for this limitation lies in the nature of PIEs being collectively frequent yet individually rare. Due to the individual scarcity of each PIE in natural language, pretrained language models struggle to capture their figurative meanings and associate them with the appropriate sentence contexts. Therefore, instead of increasing the amount of training data during pre-training to include a sufficient number of idiomatic sentences, which would be costly and impractical in most working situations, our method supplies explicit knowledge of the PIE's figurative meaning (i.e., their dictionary definitions) to facilitate the model's learning of PIE meanings and their association with corresponding contexts.
>
> While it would be interesting to explore the extent to which our model can generalize to unseen idioms during training, we do not anticipate strong generalizability due to the design of our method. Any generalization observed would likely be more indicative of what is already encoded in the pretrained language models, rather than the merit of our approach.
>
> -----
>
> ***Question**: For the training techniques that are not specific to fusing the two models (copy objective, infilling, cosine similarity objective) have you applied those to BART or GIEA without including the fusion attention layer? That could help to disentangle your various contributions, because running full PIER might be impractical for future work, whereas adding one of those components might be quite a bit easier. (e.g. I can imagine that the copy objective in itself could help BART improve).*
>
> Thank you for highlighting this point. Indeed, applying only similarity learning and prompt infilling objectives to fine-tune a BART model without the attention fusion layer is an important and informative baseline. Consequently, we have incorporated this as a new baseline model, named "BART-FT," in our experimental results, which we will include in the paper.
>
> As demonstrated in the augmented tables below, the BART-FT model exhibits similar intrinsic quality to GIEA or BART and underperforms PIER+ in all intrinsic evaluation tasks and PIE processing tasks (i.e., SenseCLF and SpanDET). However, it performs competitively with all other models in the NLU tasks (i.e., SentCLF and ParaID). This outcome indicates that our core architectural design, which involves applying attention fusion to contextually combine BART and GIEA's embeddings, is essential. Merely fine-tuning BART with auxiliary learning objectives cannot produce PIE embeddings with accurate semantic meanings and robust PIE processing capabilities.
>
> **Intrinsic Evaluation**
>
> | Model   | H-Score (↑) | CosDist (↑) | DiffSim (↓) |
> | ------- | ----------- | ----------- | ----------- |
> | BART    | 0.4546      | 0.0379      | 0.7495      |
> | GIEA    | 0.6450      | 0.2284      | 0.8224      |
> | BART-FT | 0.4510      | 0.0331      | 0.8198      |
> | PIER+   | 0.6095      | 0.1838      | 0.3230      |
>
> **Extrinsic Evaluation**
>
> | Model   | SenseCLF |        | SpanDET |        | SentCLF |        | ParaID |        |
> |---------|----------|--------|---------|--------|---------|--------|--------|--------|
> |         | F1       | Acc    | SA      | TR     | F1      | Acc    | F1     | Acc    |
> | BART    | 0.9589   | 0.9371 | 0.5076  | 0.7545 | 0.9246  | 0.9232 | 0.9165 | 0.9225 |
> | GIEA    | 0.9573   | 0.9325 | 0.7601  | 0.9075 | 0.9145  | 0.9117 | 0.9046 | 0.9103 |
> | BART-FT | 0.9614   | 0.9408 | 0.472   | 0.7907 | 0.9364  | 0.9346 | 0.9152 | 0.9207 |
> | PIER+   | 0.9765   | 0.9637 | 0.793   | 0.9101 | 0.929   | 0.9278 | 0.9068 | 0.9122 |

---

### Official Review · Reviewer_mdhy · 2023-08-12

**Soundness:** 4

**Excitement:**

3: Ambivalent: It has merits (e.g., it reports state-of-the-art results, the idea is nice), but there are key weaknesses (e.g., it describes incremental work), and it can significantly benefit from another round of revision. However, I won't object to accepting it if my co-reviewers champion it.

**Missing References:**

At BERT age,
[1] there have been similar methods motivated by using both literal and idiomatic representations for downstream tasks, i.e. {tan-jiang-2020-bert,tan-jiang-2021-learning}.
[2] there have been datasets that also discussed methods on compositional representations {tayyar-madabushi-etal-2021-astitchinlanguagemodels-dataset} and other methods from SemEval 2022 Task 2.
[3] As the model does not lose NLU abilities but achieve better embeddings, I am wondering how the model performs on an idiom paraphrase identification task, which is proposed by {pershina-etal-2015-idiom} and {tan-jiang-2021-bert}.

**Paper Topic And Main Contributions:**

The paper studies noncompositionality of phrases and proposes a model PIER to generate contextualized representations for them. PIER is constructed over BART and GIEA (BART with Adapter) to address the discrepency that GIEA's representations for the idiomatical and literal PIE are highly correlated (has a high similarity). Experiments for intrinsic evaluation show that the new method is effective in separate literal and idiomatic embeddings. Extrinsic evaluations indicate that PIER can achieve significant improvement on PIE-realted tasks.

**Reasons To Accept:**

The paper proposes a unified view on learning PIE representations, it integrates several past works like BART, GIEA and MPNet. To make comparisons with these methods, the paper designs different baselines addressing possible directions of fusion.

From evaluation perspective:
[1] PIER shows consistent improvement over GIEA on generating contextually appropriate embeddings for idiomatic and literal PIEs.
[2] PIER is useful on downstream tasks without suffering its NLU capabilities.


**Reasons To Reject:**

[1] PIER works as an interpolation of BART and GIEA, its performance is somehow expected on intrinsic evaluation.
[2] PIER still face the same challenge as GIEA that the method is applicable only if a PIE is seen during training stage. Although the authors pointed out this in Limitations, it's better to explore to what extend that idiomaticity can be `sensed` by models simutaneously.

**Reproducibility:**

5: Could easily reproduce the results.

**Reviewer Confidence:**

5: Positive that my evaluation is correct. I read the paper very carefully and I am very familiar with related work.

**Typos Grammar Style And Presentation Improvements:**

[1] Line 116: duplicate full stops.
[2] Line 141: Contribution (3) is missing.

---

> ### Author Rebuttal · Authors · 2023-08-28
>
> ***Question**: PIER works as an interpolation of BART and GIEA; its performance is somehow expected on intrinsic evaluation.*
>
> Thank you for the review. We would like to clarify that our PIER model is not a mere interpolation between BART and GIEA, as it goes beyond simple combination techniques by disambiguating idioms' literal and figurative senses based on the sentence context and generating appropriate embeddings with accurate semantics. A naïve interpolation approach, such as concatenation or averaging BART and GIEA's embeddings, would indeed result in high H-scores and CosDist in the intrinsic evaluation due to inheriting GIEA's embedding. However, this would also lead to an undesirable, high DiffSim score, which represents the difference between the embeddings for the literal and figurative meanings of the same PIE that are not clearly distinguished.
>
> As demonstrated in the table below (results were not part of the paper), averaging or concatenating BART and GIEA's embeddings results in a DiffSim score that is approximately 0.47 higher than our proposed PIER model. This indicates that our PIER model surpasses simple interpolation methods by effectively discerning contextual ambiguity, leading to superior embedding quality.
>
> | Model         | H-Score (↑) | CosDist (↑) | DiffSim (↓) |
> | ------------- | ----------- | ----------- | ----------- |
> | BART          | 0.4546      | 0.0379      | 0.7495      |
> | GIEA          | 0.6450      | 0.2284      | 0.8224      |
> | Concatenation | 0.6214      | 0.1503      | 0.7934      |
> | Averaging     | 0.6154      | 0.1355      | 0.7954      |
> | PIER+         | 0.6095      | 0.1838      | 0.3230      |
>
> To provide better clarity on this point, we will include the results from the interpolation methods in our revision and explain their significance.
>
> -----
>
> ***Question**: PIER still faces the same challenge as GIEA, that the method is applicable only if a PIE is seen during the training stage. Although the authors pointed out this in Limitations, it's better to explore to what extent that idiomaticity can be sensed by models simultaneously.*
>
> Thank you for your comment. We acknowledge that like GIEA, our proposed PIER model, is limited to working with PIEs that are encountered during training. However, we contend that this constraint is necessary to ensure high-quality representations for PIEs when using smaller-scale pretrained language models that have not adequately learned the figurative meanings of PIEs during their pre-training phase.
>
> The main reason for this limitation lies in the nature of PIEs being collectively frequent yet individually rare. Due to the individual scarcity of a large number of PIEs in natural language, pretrained language models struggle to capture their figurative meanings and to associate them with the appropriate sentence contexts. Therefore, instead of increasing the amount of training data during pre-training to include a sufficient number of idiomatic sentences, which would be costly and impractical in most working scenarios , our method supplies explicit knowledge of the PIE's figurative meaning (i.e., their dictionary definitions) to facilitate the model's learning of PIE meanings and their association with corresponding contexts.
>
> While it would be interesting to explore the extent to which our model can generalize to unseen idioms during training, we do not anticipate strong generalizability due to the design of our method (highlighted above). Any generalization observed would likely be more indicative of what is already encoded in the pretrained language models rather than the merit of our approach.

---

### Meta-Review · Area_Chair_CnaH · 2023-09-18

**Recommendation:** 3

**Metareview:**

This paper proposes a model based on BART to generate contextualized representations of non-compositional expressions (specifically, English potentially idiomatic expressions; PIEs), showing good results on PIE processing tasks. Reviewers mostly agreed that the results were sound, though the paper could benefit from more ablations/analysis demonstrating the need for certain modeling decisions. This paper represents a focused work providing a new state-of-the-art for PIE identification, which will be of interest to those in the *CL community focused on modeling idiomatic/MWE expressions, but may be less exciting to the community more broadly.

---

### Decision · Program_Chairs · 2023-10-07

**Decision:**

Accept-Findings

**Comment:**

This paper proposes a model based on BART to generate contextualized representations of non-compositional expressions (specifically, English potentially idiomatic expressions; PIEs), showing good results on PIE processing tasks. Reviewers mostly agreed that the results were sound, though the paper could benefit from more ablations/analysis demonstrating the need for certain modeling decisions. This paper represents a focused work providing a new state-of-the-art for PIE identification, which will be of interest to those in the *CL community focused on modeling idiomatic/MWE expressions, but may be less exciting to the community more broadly.